# Autoimmune Rheumatic Diseases: An Update on the Role of Atherogenic Electronegative LDL and Potential Therapeutic Strategies

**DOI:** 10.3390/jcm10091992

**Published:** 2021-05-06

**Authors:** Der-Yuan Chen, Tatsuya Sawamura, Richard A. F. Dixon, José Luis Sánchez-Quesada, Chu-Huang Chen

**Affiliations:** 1Translational Medicine Center, China Medical University Hospital, Taichung 404, Taiwan; dychen1957@gmail.com; 2Rheumatology and Immunology Center, China Medical University Hospital, Taichung 404, Taiwan; 3College of Medicine, China Medical University, Taichung 404, Taiwan; 4Department of Molecular Pathophysiology, Shinshu University School of Medicine, Matsumoto 390-8621, Japan; sawamura@shinshu-u.ac.jp; 5Department of Life Innovation, Institute for Biomedical Sciences, Shinshu University, Matsumoto 390-8621, Japan; 6Molecular Cardiology Research Laboratories, Texas Heart Institute, Houston, TX 77030, USA; RDixon@texasheart.org; 7Cardiovascular Biochemistry Group, Biomedical Research Institute IIB Sant Pau, 08041 Barcelona, Spain; jsanchezq@santpau.cat; 8CIBER of Diabetes and Metabolic Diseases (CIBERDEM), 08041 Barcelona, Spain; 9Vascular and Medicinal Research, Texas Heart Institute, Houston, TX 77030, USA; 10New York Heart Research Foundation, Mineola, NY 11501, USA

**Keywords:** electronegative LDL, L5, atherosclerotic cardiovascular disease (ASCVD), autoimmune rheumatic diseases (AIRDs), lectin-like oxidized low-density lipoprotein receptor 1 (LOX-1), interleukin 1β (IL-1β)

## Abstract

Atherosclerosis has been linked with an increased risk of atherosclerotic cardiovascular disease (ASCVD). Autoimmune rheumatic diseases (AIRDs) are associated with accelerated atherosclerosis and ASCVD. However, the mechanisms underlying the high ASCVD burden in patients with AIRDs cannot be explained only by conventional risk factors despite disease-specific factors and chronic inflammation. Nevertheless, the normal levels of plasma low-density lipoprotein (LDL) cholesterol observed in most patients with AIRDs do not exclude the possibility of increased LDL atherogenicity. By using anion-exchange chromatography, human LDL can be divided into five increasingly electronegative subfractions, L1 to L5, or into electropositive and electronegative counterparts, LDL (+) and LDL (−). Electronegative L5 and LDL (−) have similar chemical compositions and can induce adverse inflammatory reactions in vascular cells. Notably, the percentage of L5 or LDL (−) in total LDL is increased in normolipidemic patients with AIRDs. Electronegative L5 and LDL (−) are not recognized by the normal LDL receptor but instead signal through the lectin-like oxidized LDL receptor 1 (LOX-1) to activate inflammasomes involving interleukin 1β (IL-1β). Here, we describe the detailed mechanisms of AIRD-related ASCVD mediated by L5 or LDL (−) and discuss the potential targeting of LOX-1 or IL-1β signaling as new therapeutic modalities for these diseases.

## 1. Introduction

Atherosclerosis is a chronic inflammatory process that leads to vascular atheromatous plaque buildup, usually not developing into full-blown atherosclerotic cardiovascular disease (ASCVD) until later in life [1]. However, an exception is patients with autoimmune rheumatic diseases (AIRDs), who often present with accelerated ASCVD manifestations including death at a young age [2]. Rheumatoid arthritis (RA), systemic lupus erythematosus (SLE), systemic sclerosis (SSc), polymyositis (PM)/dermatomyositis (DMtis), and primary Sjögren’s syndrome (pSS) [3,4,5,6,7] are some of the AIRDs associated with accelerated ASCVD caused by inflammatory processes that result from multiorgan immune dysregulation [8]. The high ASCVD burden in these patients can be attributed to a combination of traditional risk factors, disease-specific factors, chronic inflammation, genetic components, and the use of medications [9,10,11,12,13,14,15,16,17,18]. Genetic aberrations associated with SLE-related ASCVD include variants of the genes encoding interferon regulatory factor 8 (IRF8) [19], mannose-binding lectin [20], and signal recognition particle 54-antisense 1 (SRP54-AS1) [21]. Variants of the gene encoding apolipoprotein (apo)E have been shown to be related to ASCVD risk in patients with RA [22].

Dyslipidemia usually refers to elevated levels of total cholesterol (TC), triglycerides (TG), and low-density lipoprotein cholesterol (LDL-C) and decreased levels of high-density lipoprotein cholesterol (HDL-C) [23]. Although dyslipidemia may occur in some patients with AIRDs [24,25,26,27], plasma LDL-C levels are almost universally normal in these patients at the time of admission for ASCVD complications [28,29]. LDL is a lipoprotein class that varies in density, size, electric charge, and composition among its particles. In most experimental studies of atherogenic LDL, the LDL must first undergo ex vivo oxidation to exhibit atherogenicity, whereas native LDL, such as L1, is harmless in vitro or in vivo, even at a supranormal high concentration [30]. Notably, electronegative LDL is a naturally occurring LDL that exerts potent atherogenic effects in cells and animals without undergoing ex vivo oxidation [31].

Human plasma LDL can be divided into a dichotomy of electropositive and electronegative counterparts, namely, LDL (+) and LDL (−), by using anion-exchange chromatography [32]. Alternatively, LDL can also be divided into five subfractions with increasing electronegativity, called L1 to L5 [33]. L5 and LDL (−) differ from native LDL in their physicochemical and biological characteristics, including size, density, lipid and protein composition, phospholipase activity, and apo B100 conformation [34,35]. Elevations in L5 or LDL (−) percentages in total LDL have been observed in patients with familial hypercholesterolemia [36], type 1 or type 2 diabetes mellitus (DM) [37,38,39,40], coronary artery disease [41,42], uremia [43], ischemic peripheral arterial disease [44], ST-elevation myocardial infarction (STEMI) [45,46], or acute ischemic stroke [47]. Furthermore, increases in the L5 percentage (i.e., L5%) and L5 levels are associated with the intensity of metabolic syndrome [48] and are speculated to be involved in the pathogenesis of ASCVD in patients with severe mental illness [49].

L5 and LDL (−) are not recognized by the normal LDL receptor (LDLR) because of their electronegativity; rather, they signal through lectin-like oxidized LDL receptor 1 (LOX-1) to induce a broad spectrum of inflammatory reactions in adipose tissue [50] and vascular cells [51], as well as endothelial cell (EC) dysfunction and inflammation [52]. Accordingly, both L5 and LDL (−) stimulate inflammasome formation, which involves the participation and release of several cytokines, including interleukin 1β (IL-1β), IL-6, and IL-18 [51,53,54]. In monocytes, toll-like receptor 4 (TLR4) and cluster of differentiation 14 (CD14) mediate cytokine release promoted by LDL (−) [54,55]. Of importance, patients with AIRDs have been documented to have elevated levels of L5 or LDL (−) in circulation [56,57]. Because AIRD-related ASCVD presents a significant threat to many young patients [2,11,16,58,59], it is imperative to update the mechanistic roles of L5 and LDL (−) in the pathogenesis of AIRD-related ASCVD and the preclinical modalities that have a potential therapeutic value.

In this review, we highlight the existing evidence regarding the atherogenic roles of L5 and LDL (−) in AIRDs. Because these electronegative LDLs signal through LOX-1, the potential therapeutic implications of targeting LOX-1 are also discussed. We searched the MEDLINE database using the PubMed interface and reviewed the literature published in the English language from 1988 up to 30 November 2020. The keywords searched for this review included atherosclerosis, atherogenesis, vascular inflammation, ASCVD, LDL-C, oxidized LDL, electronegative LDL, LDL (−), L5 LDL, LOX-1, inflammasomes, proinflammatory cytokines, IL-1β, AIRDs, SLE, SSc, PM/DMtis, pSS, therapeutic strategies, cytokine-targeting agents, LOX-1-targeting therapy, proprotein convertase subtilisin/kexin type 9 (PCSK9), and microRNAs. The relevant drugs included statins, corticosteroids, nonsteroidal anti-inflammatory drugs (NSAIDs), and conventional synthetic disease-modifying antirheumatic drugs (csDMARDs).

## 2. Epidemiology of ASCVD in Patients with AIRDs

Cumulative evidence has indicated that the rate of subclinical atherosclerosis is higher in patients with AIRDs than in age- and sex-matched healthy control individuals [8,60,61,62]. AIRD-related ASCVD [4] has been documented in patients with RA [63,64], SLE [28,65,66], SSc [67], PM/DMtis [68], and pSS [69,70]. If ASCVD is present, the mortality rate is significantly higher in patients with AIRDs than in those without AIRDs [71,72]. ASCVD is the leading cause of mortality in patients with RA [73]. SLE-associated mortality was originally believed to appear in a bimodal pattern, with an early peak attributable to active lupus and a later peak attributable to ASCVD [5,74]. The bimodal pattern was later flattened after Cervera et al. [75] reported that ASCVD could occur at any time during the disease course. These findings strongly indicate the importance of early screening and intervention for ASCVD in patients with AIRDs.

## 3. Electronegative LDL and Its Atherogenicity

Freshly isolated human plasma L5 and LDL (−) are highly atherogenic. They are treated to prevent oxidation during isolation, affirming that they are the naturally occurring LDL culprits responsible for atherosclerosis development and ASCVD progression.

### 3.1. The Atherogenic Mechanisms of Electronegative LDL

L5 was isolated for the first time from human plasma in 2003 [33,76] and was shown to inhibit the transcription of the pro-survival gene fibroblast growth factor-2 (*FGF-2*) in arterial ECs through an Akt-mediated autoregulatory loop, thereby disrupting mitochondrial stability and inducing apoptosis [76]. Furthermore, neutralizing the PAF receptor (PAFR) significantly attenuated the L5-induced downregulation of *FGF2* and the concomitant apoptosis of ECs, suggesting that L5 signals in part through PAFR [76,77]. Although L5 is not an oxidized LDL (oxLDL) entity, it shares many functional properties with experimentally prepared oxLDL. Similarly, to oxLDL, L5 stimulates the p53-dependent activation of the proapoptotic protein Bax, leading to the apoptosis of differentiated endothelial progenitor cells (EPCs) [78]. L5, which is increased in the plasma of chronic smokers, impairs EPC differentiation by inhibiting Akt phosphorylation via LOX-1 [31].

The atherogenicity of L5 can be attributed to the multiple stress pathways it induces in vascular ECs [79]. By downregulating the expression of the endoplasmic reticulum (ER) chaperone proteins ORP150, Grp94, and Grp58, L5 induces ER stress by suppressing mitochondrial proteins Prdx3 and ATP synthase and decreases the mitochondrial membrane potential to result in mitochondrial dysfunction [79]. Additionally, L5 upregulates heterogeneous nuclear ribonucleoproteins (hnRNP) C1/C2 to promote macrophage recruitment and monocyte chemoattractant protein 1 (MCP-1) to accelerate inflammation [79]. Of importance, L5 upregulates vascular endothelial growth factor (VEGF) to facilitate inflammatory angiogenesis and downregulates *FGF-2* transcription to inhibit compensatory angiogenesis [76,79,80,81]. Increased plasma C-reactive protein (CRP) levels are associated with the occurrence and severity of acute coronary syndrome [82]. By augmenting CRP expression in vascular ECs, L5 further intensifies its proinflammatory property [83].

In ECs, L5 selectively downregulates mitochondria-stabilizing Bcl-2 and upregulates mitochondria-destabilizing Bax and Bad proteins [84]. Exposure of differentiated EPCs to L5 also results in an increased transfer of mitochondria-derived superoxide anion to p53, which stimulates a conformational change in Bax that promotes its translocation to the mitochondria, resulting in the apoptosis of these cells [78].

Daily intravenous injection of human L5 into C57B6/J mice during a course of four weeks can induce aortic endothelial senescence accompanied by γH2AX deposition and DNA damage [85]. In cultured human aortic ECs, L5 can augment mitochondrial oxygen consumption and free radical production, leading to ATM activation, nuclear γH2AX deposition, Chk2 phosphorylation, and TP53 stabilization. L5 also decreases human telomerase reverse transcriptase (hTERT) protein levels and activity. In conclusion, L5 may promote mitochondrial free radical production and activate the DNA damage response to induce premature vascular endothelial senescence that leads to atherosclerosis.

An alternative signaling pathway of LDL (−) in monocytes and macrophages is through its binding to the TLR4/CD14 complex [55]. The binding of LDL (−) to CD14 signals the TLR4 and PI3K/Akt pathway, leading to the activation of nuclear factor-kB (NF-kB), activator protein-1 (AP-1), and cAMP response element-binding protein (CREB) nuclear transcription factors and the subsequent release of numerous cytokines [54]. In this context, LDL (−) activates the inflammasome, acting as a priming signal for IL-1β secretion and inducing the expression of pro-IL-1β and NLRP3 receptor [53].

### 3.2. The Assocation of Electronegative LDL with ASCVD in Patients with AIRDs

In three comprehensive studies, researchers have recently shown that circulating L5 levels are significantly increased in patients with RA or SLE whose plasma LDL-C levels are equal to or lower than those of healthy control individuals [56,57,86]. In a cohort of patients with RA, the plasma L5% and L5 concentration were significantly higher in those with subclinical atherosclerosis than in those without it, despite similar LDL-C levels among all the study participants [56]. The plasma L5% and L5 concentration were also positively correlated with the extent of carotid artery atherosclerosis reflected by the intima–media thickness (IMT), disease activity score, and 10-year ASCVD risk score [56]. Further investigation revealed that L5 contributes to atherogenesis by augmenting macrophage foam cell formation, upregulating integrin CD11c expression, and enhancing inflammatory mediators including IL-6, IL-8, and tumor necrosis factor (TNF)-α [86]. These findings provide a mechanistic link between L5 and ASCVD development and clinical manifestations [56,57,79,86]. Most importantly, both L5% and L5 concentration declined significantly after six months of DMARD therapy in RA patients [56], suggesting the dynamic and reversible nature of L5 formation. SLE is a systemic vascular disease that mostly affects young females [87]. The rapid progression of ASCVD in patients with SLE often leads to accelerated vascular aging and death that cannot be explained by conventional risk stratification in this age-and-sex group [2,28,65]. Similar to the scenario of patients with RA, patients with SLE also have a significantly higher L5% and L5 concentration in circulation than do healthy control individuals despite patients with SLE having a plasma LDL-C level lower than that of control individuals [57]. In patients with SLE, increased L5% correlates positively with mean blood pressure, IMT, pulse wave velocity, and blood levels of CD16+ monocytes and CX3CL1 cytokines [57]. Both L5 and the total LDL of patients with SLE have a high content of lysophosphatidylcholine (LPC) and platelet-activating factor (PAF). When administered to young *ApoE^−/−^* mice, L5 and total LDL of patients with SLE can induce increases in plasma CX3CL1, aortic fatty-streak areas, aortic vascular aging, and macrophage infiltration into the aortic wall [57]. As expected, LPC exerts similar effects in vivo, whereas LDL from control individuals has negligible effects. In vitro, synthetic PAF and LPC can also induce CD16 expression in human monocytes and monocyte–EC adhesion, as does the lipid moiety of L5 in patients with SLE [57]. Thus, plasma L5 levels are increased in both patients with RA and those with SLE, thereby contributing to their accelerated atherosclerosis. Increased plasma L5 levels are also associated with an increased risk of ASCVD in patients with various other clinical diagnoses, including patients with uremia [43], ischemic peripheral arterial disease [44], or asymptomatic type 2 DM [39,40]. The risk of ASCVD in patients with asymptomatic type 2 DM is equivalent to that in patients with RA [88].

### 3.3. Combined Atherogenicity of Electronegative VLDL and LDL

Triglyceride-rich lipoproteins, very low-density lipoprotein (VLDL) in particular, may exert atherogenic effects both directly and as a precursor of LDL. VLDL can also be chromatographically divided into subfractions with increasing electronegativity (V1–V5); the most electronegative subfraction, V5, is more toxic in ECs than L5 is at an equal concentration [89]. Notably, the combined electronegativity of L5 and V5 is significantly greater in individuals with a high risk of ASCVD than in those with a low risk of CHD [42]. In *ApoE^−/−^* mice, the combined increased electronegativity of LDL and VLDL is accompanied by increased lipid accumulation and cellular senescence in the aorta, further substantiating their contribution to ASCVD [42].

## 4. Role of LOX-1 in Electronegative LDL Signaling

Liquid chromatography/mass spectrometry (LC/MS^E^) analysis revealed that L1′s protein framework is composed of 99% apoB100 with trace amounts of other proteins, whereas L5 contains 60% apoB100 with higher amounts of apo(a), apoE, apoAI, and apoCIII than does L1. In contrast to L1, which has an isoelectric point (pI) of 6.620, L5 has a pI of 5.5 or less, which contributes to its electronegativity, rendering it unrecognizable by LDLR [90]. LOX-1, originally discovered by Sawamura et al. [91] in 1997, is a 50-kDa type II membrane protein of the C-type lectin family. An important scavenger receptor, LOX-1 exhibits binding activity for multiple ligands, including oxidized LDL (ox-LDL), polyanionic chemicals, anionic phospholipids, cellular ligands (apoptotic/aged cells, activated platelets, and bacteria), and bile salt-dependent lipase [92,93]. After being internalized via LOX-1, L5 exerts its signaling in vascular ECs and monocytes [50,51,52,53,54,84,90].

In *ApoE^−/−^* mice, the specific overexpression of LOX-1 in the endothelium promotes atherosclerosis and inflammation [94]. High-cholesterol diet-induced plaque formation and proinflammatory signals can be reduced by deleting the *LOX-1* gene [95]. LOX-1 is also a sensor of danger signals [96] involved in reactive oxygen species generation, which leads to NLRP3 inflammasome activation [97]. In addition, the binding of L5 or LDL (−) to LOX-1 leads to intracellular signaling and activates multiple downstream events that are critical steps in atherosclerosis. Recent studies have revealed that LDL (−) from patients with STEMI could induce the production of IL-1β, granulocyte colony-stimulating factor (G-CSF), and granulocyte/monocyte (GM)-CSF through a LOX-1-dependent pathway [98,99] and that the atherogenic and proinflammatory responses elicited by LDL (−) could be offset by blocking LOX-1 with a neutralizing antibody or shRNA knockdown [98]. Moreover, increased LOX-1 expression has been observed in human atherosclerotic lesions and in experimental animal models [100,101], supporting a crucial role of LOX-1 in atherosclerosis.

### 4.1. The Potential Atherogenic Role of LOX-1 in Patients with AIRDs

The evidence that LOX-1 plays a role in AIRD-associated ASCVD is substantiated by the finding that plasma from patients with RA can upregulate LOX-1 expression in human macrophages [102]. In addition, CRP can enhance endothelial LOX-1 expression [103]. CRP and LOX-1 may form a positive feedback loop with L5 in atherogenesis, whereby increased levels of atherogenic LDL in patients with cardiovascular risks induce ECs to express CRP, which may in turn increase the expression of LOX-1 to promote the uptake of electronegative LDL into ECs [104].

A recent study confirmed that plasma L5 but not L1 from patients with both RA and subclinical atherosclerosis can enhance LOX-1 expression in macrophages [56]. In addition, LOX-1 expression levels in patients with RA positively correlate with plasma L5% or L5 concentrations and are significantly associated with 10-year ASCVD risk scores [56]. Another study showed that perturbations in plasma lipid content can activate LOX-1 expression and promote inflammatory responses, suggesting that LOX-1 is a potential driver of ASCVD risk in patients with SLE [105]. LOX-1 expression can be upregulated by other inflammatory mediators [84], and the binding of L5 to LOX-1 leads to the feedforward elevation of LOX-1 expression [56], further promoting subsequent atherogenic and inflammatory responses.

### 4.2. Atherogenic Signaling of Electronegative LDL and LOX-1 in Patients with AIRDs

A schematic summary of the pathogenic role of electronegative LDL and LOX-1 signaling in AIRD-related ASCVD is shown in Figure 1. The emergence of ASCVD in patients with AIRDs is believed to result from complex interactions among traditional cardiovascular risk factors, disease-specific factors, disease-related inflammatory mediators, and the medications used. Increased levels of electronegative LDL in patients with diabetes [37,38,39] may be produced by lipolysis, generating an increased level of non-esterified fatty acid (NEFA) [33]. Cigarette smoking promotes a significant increase in the production of L5, contributing to smoking-associated ASCVD via LOX-1 [31,106]. Moreover, rabbits fed an atherogenic diet were shown to produce highly proinflammatory LDL (−) [107]. The high affinity of LDL (−) for LOX-1 but low affinity for LDLR may be explained by its high NEFA content [108], its high degree of aggregation [109], and the presence of an abnormal conformation of apoB100 [110]. The L5 and LDL (−) signaling pathways mediated by LOX-1 provide further insight into the pathogenesis of ASCVD and may represent a therapeutic target for patients with AIRDs.

## 5. Proposed Therapeutic Strategies for Targeting ASCVD in Patients with AIRDs

Epidemiologic studies have shown increased morbidity and mortality rates due to ASCVD in patients with AIRDs [4,39,40,41,42,43,44,45,46,47,51], suggesting the importance of screening and preventive strategies for ASCVD, as well as its management. Although there is a paucity of consensus or randomized studies regarding therapeutic strategies for ASCVD in AIRDs, both the American College of Rheumatology (ACR) and the European League Against Rheumatism (EULAR) recommendations emphasize that patients with AIRDs should undergo risk factor monitoring, CVD risk modification, and dyslipidemia therapy, if needed [111,112].

### 5.1. Modifications of Traditional CVD Risk Factors

Given that cigarette smoking is implicated in L5–LOX-1 signaling pathway-related atherosclerosis [31,106] and that smoking cessation could reduce an independent ASCVD risk marker (i.e., the LOX index) [113], smoking cessation should be a priority. Avoidance of intense aerobic exercise is recommended because it may increase the NEFA content of LDL (−) [114]. Considering that an atherogenic diet induces LDL (−)-related inflammation [107], replacing it with the Mediterranean diet may reduce ASCVD risk in patients with AIRDs [115].

### 5.2. Tight Control of Disease Activity and the Adjustment of Medications

Given that AIRD-related immune dysregulation has an important role in premature atherosclerosis, the control of disease activity is essential. In patients with AIRDs, ASCVD risk is significantly associated with acute-phase reactants, inflammatory cytokines, autoantibody positivity, and specific T cell subsets [116,117]. The acute-phase reactant CRP in patients with AIRDs can enhance endothelial LOX-1 expression [103]. Chang et al. [56] also revealed a relationship between the plasma L5 proportion and the RA disease activity. The inflammatory mechanisms in AIRDs may intensify oxidative stress, triggering a broad range of proatherogenic lipid changes [118]. According to the EULAR guidelines [111,112], adequate control of disease activity reduces the risk of ASCVD in patients with AIRDs. Given the pathogenic role of proinflammatory cytokines in AIRD-related atherosclerosis, tight control of disease activity with biologic DMARDs, including TNF-α inhibitors, IL-6 inhibitors, and IL-17 inhibitors, may reduce the risk of ASCVD [119,120,121].

Increasing evidence has indicated that corticosteroids are associated with hypertension and dyslipidemia [122] and that nonsteroidal NSAIDs may adversely affect cardiovascular outcomes [123]. The EULAR guidelines recommend using the lowest dose of corticosteroids and prescribing NSAIDs with caution [111,112]. In contrast, recent studies revealed a beneficial effect of hydroxychloroquine (HCQ) on cardiovascular outcomes in patients with RA [124,125]. Other csDMARDs, such as methotrexate, could also reduce ASCVD risk [126].

### 5.3. Lipid-Lowering Therapy

Although cardiovascular interventional trials have not been performed specifically in patients with AIRDs, statins are the preferred therapeutic agent for dyslipidemia. Statin therapy could reduce the proportion of LDL (−) and attenuate inflammation in patients with hyperlipidemia [127,128]. Simvastatin therapy improves the affinity of LDL (−) for the LDLR in parallel with a decrease in the proportion of LDL (−) [127]. Given a bidirectional link between LOX-1 and PCSK9, especially in an inflammatory status [129], a PCSK9 inhibitor could reduce LOX-1 expression. L5 is a promising marker for predicting ASCVD [56,57,86]. The therapeutic goal of controlling L5 should be to lower its plasma concentration below 1.7 mg/dL. If the L5 concentration is higher than 1.7 mg/dL, a lipid-lowering treatment, such as statins or PCSK9 inhibitors, may be required regardless of the absolute LDL level [130].

### 5.4. Potential Therapeutic Strategy Targeting Electronegative LDL and LOX-1 Pathway

Given that LOX-1 is a multiligand and multifunctional receptor underlying cardiovascular dysfunction and ASCVD [92,93,94,95,96,105], a therapy targeting LOX-1 is promising as a treatment for atherosclerosis and vasculopathy [131]. MicroRNAs, which are short noncoding RNAs, have been recently identified as immune regulators that post-transcriptionally repress mRNA expression. Recent studies have revealed that microRNA let-7g and microRNA-98 may target LOX-1 as a therapy for atherosclerosis [132,133]. In addition, the transcription factor NF-κB is involved in the development and progression of both vascular inflammation and ASCVD [134]. Interestingly, NF-κB and sirletin-1 feature the characteristics of antagonistic crosstalk [135]. Sirletin-1 drives anti-inflammatory responses and enhances the resolution of vascular inflammation. Based on the findings of previous studies, meta-analyses, reviews, recommendations, and our results, we propose a potential therapeutic strategy with an emphasis on targeting the L5/LDL (−) and LOX-1 pathway in patients with AIRDs (Figure 2).

## 6. Conclusions

Accelerated atherosclerosis with an increased risk of ASCVD is well-documented in patients with AIRDs. Furthermore, cumulative evidence has indicated that the L5/LDL (−) and LOX-1 signaling pathway contributes to atherogenicity. We propose probable mechanisms behind this signaling in the pathogenesis of AIRD-related ASCVD and a potential therapeutic strategy targeting L5/LDL (−) and LOX-1 signaling. Regularly updating these pathologic mechanisms and therapeutic strategies will be critical as new evidence emerges.

## Figures and Tables

**Figure 1 jcm-10-01992-f001:**
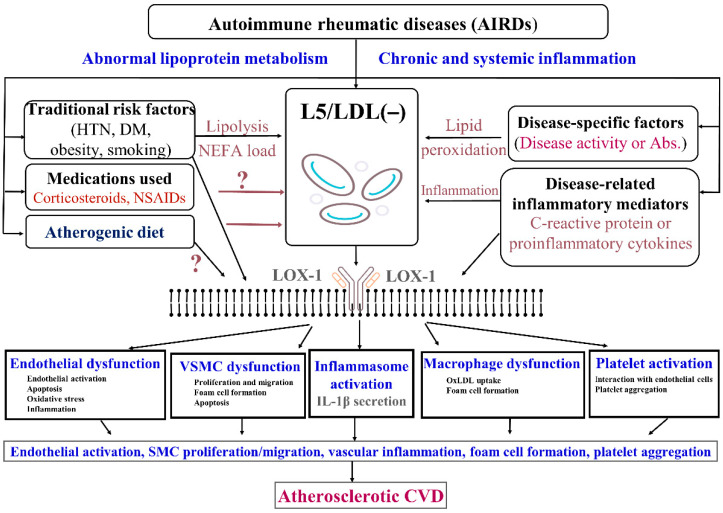
The probable atherogenic role of electronegative LDL (LDL(−)/L5) in patients with AIRDs. The increased levels of LDL (−) may be produced by lipolysis with non-esterified fatty acid (NEFA) loading in patients with diabetes. Cigarette smoking is associated with L5 formation and LOX-1-induced atherosclerosis. C-reactive protein induces LOX-1 expression. L5/LDL (−) and the LOX-1 signaling pathway lead to atherogenesis, vascular inflammation, and platelet activation/aggregation. LDL: low-density lipoprotein; LOX-1: lectin-like oxidized low-density lipoprotein receptor 1; HTN: hypertension; DM: diabetes mellitus; NSAIDs: nonsteroidal anti-inflammatory drugs; Abs.: autoantibodies; oxLDL: oxidized LDL; IL-1β: interleukin 1β; CVD: cardiovascular disease.

**Figure 2 jcm-10-01992-f002:**
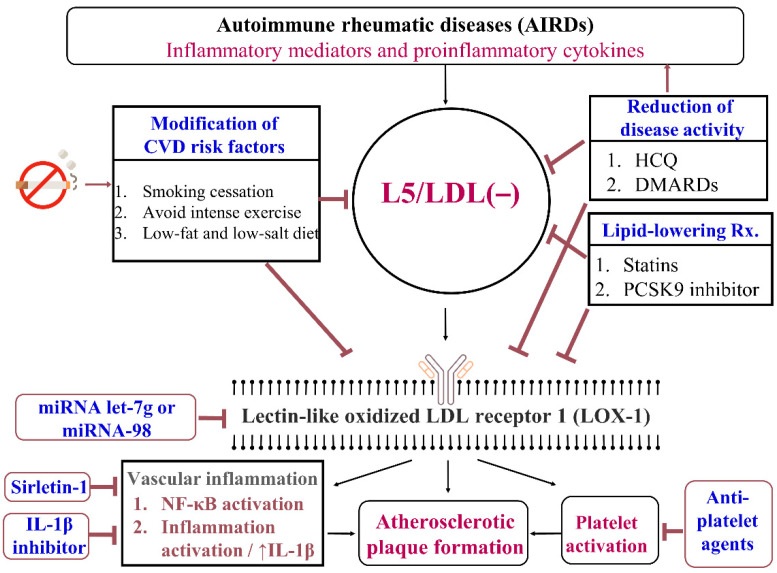
A potential therapeutic strategy for the targeting L5/LDL(−) and LOX-1 pathway in patients with AIRDs. Major therapeutic strategies include modifications of the traditional CVD risk factors; control of AIRD activity; and the use of lipid-lowering agents, cytokine-targeting agents, and LOX-1-targeting therapy. CVD: cardiovascular disease; LDL: low-density lipoprotein; HCQ: hydroxychloroquine; DMARDs: disease-modifying anti-rheumatic drugs; PCSK9: proprotein convertase subtilisin/kexin type 9; miRNA: microRNA; NF-kB: nuclear factor-kB; IL-1β: interleukin 1β.

## Data Availability

Not applicable.

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
