# Peer review of "Autoimmune Rheumatic Diseases: An Update on the Role of Atherogenic Electronegative LDL and Potential Therapeutic Strategies"

_jcm, 2021, doi:10.3390/jcm10091992_

Round 1
Reviewer 1 Report
I read the review “Systemic Autoimmune Rheumatic Diseases: An Update on the Role of Atherogenic Electronegative LDL and Potential Therapeutic Strategies” and I have found it interesting and well written. I have only few minor comments.
- please replace SRADs with AIRDs (autoimmune rheumatic diseases) that is the term used for systemic autoimmune diseases (the acronym SARDs is also for “sudden acquired retinal degeneration syndrome”).
- I suggest to anticipate the paragraph 3.3 before the paragraphs 3.1 and 3.2 (better explain the atherogenic mechanisms of electronegative LDL first).
- In the Figure 1, under “Disease-specific factors” I suggest replacing “activity scores” with “disease activity” (express the same concept but avoiding the readers asking what scores).
Author Response
We thank the Reviewer for the positive and constructive comments. We have replaced SARDs with AIRDs throughout the revised manuscript. In addition, we have placed material from paragraph 3.3 so that it is presented before paragraphs 3.1 and 3.2 (page 3, line 123). As recommended by the Reviewer, we have also replaced “activity scores” with “disease activity” in Figure 1.
Reviewer 2 Report
This is a comprehensive review regarding the role of L5 in the pathogenesis of atherosclrerosis in the setting of systemic autoimmunity. Given that genetic players have been recently shown to contribute to atherosclerosis related to RA and lupus, it would be desirable to include these studies in the introduction section suggesting if any potential interactions with heightened L5 levels.
Author Response
We thank the Reviewer for this comment. We have added additional references and mention of the genetic players in the Introduction on page 2, line 53, as follows: “The high ASCVD burden in these patients can be attributed to a combination of traditional risk factors, disease-specific factors, chronic inflammation, genetic components, and the use of medications [9-18]. Genetic aberrations associated with SLE-related ASCVD include variantsof the genes encoding interferon regulatory factor 8 (IRF8) [19], man-nose-binding lectin [20], and signal recognition particle 54-antisense 1 (SRP54-AS1) [21]. Variants of the gene encoding apolipoprotein (apo)E have been shown to be related to ASCVD risk in patients with RA [ 22].”
- Mahmoudi M, Aslani S, Fadaei R, et al. New insights to the mechanisms underlying atherosclerosis in rheumatoid arthritis. Int J Rheum Dis 2017, 20, 287-297.
- Nezosa A, Evangelopoulosb ME, Clio P. Mavragani CP. Genetic contributors and soluble mediators in prediction of autoimmune comorbidity. J Autoimmun 2019, 104, 102317.
- Leonard D, Svenungsson E, Sandling JK, et al. Coronary aheart disease in systemic lupus erythematosus is associated with interferon regulatory factor-8 variants. Circulation Cardiovascular Genetics 2013, 6,255-263.
- Ohlenschlaeger T, Garred P, Madsen HO, et al. Mannose-binding lectin variant alleles and the risk of arterial thrombosis in systemic lupus erythematosus. N Engl J Med 2004, 351, 260-267.
- Leonard D, Svenungsson E, Dahlqvist J, et al. Novel gene variants associated with cardiovascular disease in systemic lupus erythematosus and rheumatoid arthritis. Ann Rheum Dis 2018, 77, 1063-1069.
- Maehlen MT, Provan SA, de Rooy D, et al. Associations between APOE genotypes and disease susceptibility, joint damage and lipid levels in patients with rheumatoid arthritis. PLoS ONE 2013; 8: e60970.
Reviewer 3 Report
The main question addressed by the paper is the role of atherogenic electronegative LDL and potential therapeutic options
The topic is relevant and interesting.The authors presented an interesting original review article described the detailed mechanisms of SARDs-related ASCVD mediated by L5 or LDL (-). The paper is well written and easy to read.
The manuscript's strengths is an interesting discussion about the potential targeting of Lox-1 or IL-1 signaling as new therapeutic modalities for SARDs-related ASCVD.
However, I have some minor comments about this paper.
I believe that Section 5.2. (Tight Control of Disease Activity and the Adjustment of Medications) should be extended to influence biological medicines (eg TNFI, IL-6i, IL-17i) and inhibitors as a risk of ASCVD.
Referring to Fig. 2, a short explanation of the role of Sirletin-1 in the NF-KB activation would be useful in the text.
Author Response
We thank the Reviewer for these comments. We have addressed the Reviewer’s concern about section 5.2 on page 7, line 317 by adding the following text and references: “Given the pathogenic role of proinflammatory cytokines in AIRDs-related atherosclerosis, tight control of disease activity with biologic DMARDs, including TNF-α inhibitors, IL-6 inhibitors, and IL-17 inhibitors, may reduce the risk of ASCVD [119-121].”
- Westlake SL, et al. Tumour necrosis factor antagonists and the risk of cardiovascular disease in patients with rheumatoid arthritis: a systematic literature review. Rheumatology (Oxford) 2011, 50:518-31.
- Ait-Oufella H, Libby P, Tedgui A. Anti-cytokine immune therapy and atherothrombotic cardiovascular risk. Arterioscler Thromb Vasc Biol 2019, 39, 1510-1519.
- Ursini F, Ruscitti P, Caio GPI, et al. The effect of non-YNF-targeted biologics on vascular dysfunction in rheumatoid arthritis: a systemic literature review. Autoimmu Rev 2019, 18, 501-509.
In addition, we have added a short explanation of the role of sireletin-1 in NF-kB activation on page 8, line 345, as follows: “In addition, the transcription factor NF-κB is involved in the development and progression of both vascular inflammation and ASCVD [134]. Interestingly, NF-κB and sirletin-1 feature the characteristics of antagonistic crosstalk [135]. Sirletin-1 drives anti-inflammatory responses and enhances the resolution of vascular inflammation.”
- Fiordelisi A, Iaccarino G, Morisco C, Coscioni E, Sorriento D. NFkappaB is a key player in the crosstalk between inflammation and cardiovascular diseases. Int J Mol Sci. 2019, 20,1599.
- Kauppinen A, Suueonen T, Ojala J, Kaarniranta K, Salminen A. Antagonistic crosstalk between NF-κB and SIRT1 in the regulation of inflammation and metabolic disorders. Cellular Signaling 2013, 25, 1939-1948.